# Activation of CD14+ Monocytes via the IFN-γ Signaling Pathway Is Associated with Immune-Related Adverse Events in Hepatocellular Carcinoma Patients Receiving PD-1 Inhibition Combination Therapy

**DOI:** 10.3390/biomedicines12061140

**Published:** 2024-05-21

**Authors:** Yaoru Song, Shida Pan, Jiahe Tian, Yingying Yu, Siyu Wang, Qin Qiu, Yingjuan Shen, Luo Yang, Xiaomeng Liu, Junqing Luan, Yilin Wang, Jianing Wang, Xing Fan, Fanping Meng, Fu-Sheng Wang

**Affiliations:** 1Medical School of Chinese PLA, Beijing 100853, China; syr19960605@163.com (Y.S.); 83526542@163.com (Y.W.); 2Department of Infectious Diseases, The Fifth Medical Centre of Chinese PLA General Hospital, National Clinical Research Centre for Infectious Diseases, Beijing 100853, China; wangsiyu_1982521@163.com (S.W.); qiuqinwangqiu@163.com (Q.Q.); yingying310310@163.com (Y.S.); xyinbj302@163.com (L.Y.); liuxiaomeng224@163.com (X.L.); ljq94630@163.com (J.L.); fanxing302@aliyun.com (X.F.); 3Beijing Ditan Hospital, Capital Medical University, Beijing 100015, China; posda_pg@163.com; 4Peking University 302 Clinical Medical School, Beijing 100191, China; tianjh9888@163.com (J.T.); wjianing1001@163.com (J.W.); 5The First Affiliated Hospital of USTC, Division of Life Sciences and Medicine, University of Science and Technology of China, Hefei 230001, China; yuyingying1020@163.com

**Keywords:** hepatocellular carcinoma, immune-related adverse events, checkpoint inhibitor, monocytes, IFN-γ signaling

## Abstract

(1) Background: Immune-related adverse events (irAEs) are a series of unique organ-specific inflammatory toxicities observed in patients with hepatocellular carcinoma (HCC) undergoing PD-1 inhibition combination therapy. The specific underlying mechanisms remain unclear. (2) Methods: We recruited 71 patients with HCC undergoing PD-1 inhibition combination therapy. These patients were then divided into two groups based on irAE occurrence: 34 had irAEs and 37 did not. Using Olink proteomics, we analyzed the aberrant inflammation-related proteins (IRPs) in these patient groups. For single-cell RNA sequencing (scRNA-seq) analysis, we collected peripheral blood mononuclear cells (PBMCs) from two representative patients at the pretreatment, irAE occurrence, and resolution stages. (3) Results: Our study revealed distinct plasma protein signatures in HCC patients experiencing irAEs after PD-1 inhibition combination therapy. We clarified the relationship between monocyte activation and irAEs, identified a strongly associated CD14-MC-CCL3 monocyte subset, and explored the role of the IFN-γ signaling pathway in monocyte activation during irAEs. (4) Conclusions: The activation of monocytes induced by the IFN-γ signaling pathway is an important mechanism underlying the occurrence of irAEs in HCC patients receiving PD-1 inhibition combination therapy.

## 1. Introduction

Hepatocellular carcinoma (HCC) ranks as the sixth most prevalent cancer and the fourth leading cause of cancer-related deaths globally [1]. The mortality rate associated with liver cancer is projected to rise steadily worldwide. Based on a 2019 projection report by the World Health Organization, it is expected that over one million people worldwide will lose their lives to liver cancer by 2030 [2]. However, due to the remarkable regenerative and compensatory capabilities of the hepatic system, HCC often presents no distinguishable symptoms in its early stages. As a result, approximately 70% of patients are diagnosed at advanced stages where surgical interventions are limited, leading to an unfavorable prognosis [3].

Among the various cancer immunotherapies, ICIs targeting molecules such as PD-1 and cytotoxic T-lymphocyte-associated protein-4 (CTLA4) have emerged as a promising approach [3]. In the realm of advanced HCC, the efficacy of PD-1 ICI monotherapy remains modest, achieving objective response rates (ORRs) of only 15% to 18.3% in late-stage patients [4,5]. It is worth noting that 20% of these patients experience treatment-induced irAEs. Combination immunotherapies for HCC have shown promising results with higher ORRs, although they are accompanied by an increase in irAEs. In the CheckMate040 trial, for instance, the combination of anti-PD-1 and anti-CTLA4 therapies achieved impressive ORRs of 31% in patients with advanced HCC. However, this treatment regimen was also associated with grade 3/4 irAEs in 37% of the cases [6]. Similarly, the IMBRAVE150 trial demonstrated that the combination of anti-programmed death-ligand 1 (PD-L1) and anti-vascular endothelial growth factor A achieved ORRs of 27.3%. However, this therapeutic approach also caused grade 3/4 adverse events in 56.5% of the patients [7]. Current research indicates that the main mechanisms by which PD-1 ICI induces irAEs encompass elevated T-cell activation and proliferation, facilitation of diverse cytokine secretions, and augmentation of humoral autoimmunity. Nevertheless, the exact underlying mechanisms still need to be further clarified [8,9].

Given these findings, it is imperative to gain a deeper insight into the mechanisms underlying irAEs to improve the outcomes of immune checkpoint blockade (ICB) in HCC. The emergence of Olink proteomics and single-cell sequencing technologies enables comprehensive analyses of the immune landscape, thereby facilitating the discovery of predictive immune signatures and biomarkers [10,11].

In this study, we employed Olink proteomics and single-cell sequencing technologies to conduct a thorough examination of the serological features and immunological mechanisms linked to irAEs in HCC patients undergoing anti-PD-1 therapy.

## 2. Materials and Methods

### 2.1. Study Design and Sample Collection

From April 2020 to June 2023, 71 patients with HCC who had received PD-1 ICIs combined with other targeted therapy drugs and provided written informed consent were recruited. All the patients received treatment at the Department of Infectious Diseases, Fifth Medical Center of the PLA General Hospital. For the administration of immunotherapy drugs, sintilimab, tislelizumab, and camrelizumab were given at fixed doses of 200 mg each, while toripalimab was administered at a slightly higher dose of 240 mg. All the immunotherapies followed a three-week treatment cycle. In terms of targeted therapy, sorafenib was prescribed at a daily dose of 400 mg. Lenvatinib dosage ranged from 8 to 12 mg daily, and it was adjusted according to the patient’s body weight. Bevacizumab, however, was dosed at 15 mg per kilogram of body weight per day. These dosing regimens were carefully determined to ensure the optimal therapeutic effect while minimizing potential side effects in cancer patients undergoing treatment. The treatment plan for patients meeting the TACE treatment criteria was jointly decided by the patient and the doctor before or after systemic combination therapy. Peripheral blood samples were obtained from the patients at baseline and during treatment. Immediately following collection, plasma samples were carefully stored at −80 °C until the time of analysis to preserve their integrity and prevent any biological or chemical changes that might occur. Meanwhile, peripheral blood mononuclear cells (PBMCs) were separated from the whole blood using Ficoll–Hypaque (MD Pacific Biotechnology in Tianjin, China) density gradient centrifugation.

### 2.2. Evaluation Reference Criteria

In this study, irAEs were documented according to Version 5 of the Common Terminology Criteria for Adverse Events (CTCAE) [12]. Furthermore, the efficacy of the treatment was evaluated in 71 enrolled patients according to the modified Response Evaluation Criteria in Solid Tumors 1.1 (mRECIST 1.1) [13,14]. To assess the stage of HCC and the disease burden in patients, the Barcelona Clinic Liver Cancer (BCLC) staging system and the Model for End-Stage Liver Disease (MELD) scoring system were utilized.

### 2.3. Proteomic Profiling of Soluble Factors in Plasma

In this study, the Olink multiplexed Proximity Extension Assay (PEA) inflammation panel was utilized to quantify plasma samples from the cohort, and it encompassed 92 immune response proteins (IRPs) (Note: Olink Bioscience AB, Uppsala, Sweden). The PEA method employs pairs of oligonucleotide-labeled antibodies that bind to their target proteins. When two antibodies are in close proximity, a novel polymerase chain reaction (PCR) target sequence is formed through a proximity-dependent DNA polymerization reaction. Subsequently, standard real-time PCR techniques are employed to detect and quantify the resulting sequences. The data generated by Olink are represented as normalized protein expression values whose distribution resembles log2-transformed protein concentrations. Among our 181 samples, two samples failed the quality control or deviated from the target population and were, thus, excluded from the analysis. Additionally, 19 proteins had more than 90% of their values below the detection limit and were, therefore, also excluded from further analysis. This study evaluated 73 IRPs from 71 individuals representing different stages of irAEs. Further information on the Olink assay is available at http://www.olink.com, accessed on 15 February 2024.

### 2.4. Single-Cell RNA Preparation and Sequencing

We processed all the blood samples within two hours of collection. We used Ficoll–Plaque medium to isolate peripheral blood mononuclear cells (PBMCs). The cell survival rate was determined to be over 90% through trypan blue staining. According to the manufacturer’s specifications, we used Chromium Next GEM Single Cell 3′ v3 (10× Genomics) in Beijing, China for single-cell capture and library construction. Sequencing was performed on the Illumina NovaSeq 6000 platform.

### 2.5. Single-Cell RNA Sequencing Data Analysis

We processed single-cell expression data using the Cell Ranger pipeline (v3.0.1, 10× Genomics), which included demultiplexing, genome alignment (GRCh38), barcode counting, and unique molecular identifier (UMI) counting. We utilized Seurat (V5.0.1) for the analysis of the gene–barcode matrix and data integration of UMI counts. Subsequently, the output was imported into Seurat (V5.0.1) in R for subsequent quality control and downstream analysis of the single-cell RNA sequencing data. To calculate mitochondrial gene expression, we employed the PercentFeatureSet function from the Seurat package. As part of the quality control, we removed cells with a mitochondrial genome UMI percentage exceeding 10%. Following this, the combined matrix was scaled, and uniform manifold approximation and projection (UMAP) visualization was performed using the top 30 dimensions from principal component analysis (PCA). We applied the same scaling, dimension reduction, and clustering procedures to specific datasets of sub-clusters. Using the Wilcoxon rank-sum test, we identified significantly differentially expressed genes (DEGs) within each cluster by comparing them with other clusters.

### 2.6. Cellular Communication Analysis

We utilized the CellChat package within the R software (version 4.3.1) to conduct cellular communication analysis. CellChat infers biologically significant intercellular communications by assigning a probability score to each interaction and executing permutation tests. Circular plots and bubble charts were employed to visualize the communication networks and signaling pathways [15].

### 2.7. Defining Cell State Scores

We used the AUCell-R package to score the target gene sets. With reference to the relevant literature, we used MONOCYTE CHEMOTAXIS (GO:0002548) for the monocyte chemotaxis score and TYPE II INTERFERON-MEDIATED SIGNALING PATHWAY (GO:0060333) for the IFN-γ signaling pathway score.

### 2.8. Statistical Analysis

The statistical analysis was performed using R software (version 4.3.1). The categorical variables are presented as counts and percentages (%). Data normality was assessed using the Kolmogorov–Smirnov test. The continuous variables are reported as means ± standard deviations. Comparisons between groups of continuous variables were made using either the Student’s *t*-test or the Mann–Whitney U test, depending on the data distribution, while the categorical outcomes were analyzed using the chi-square test or Fisher’s exact test. Statistical significance was set at a two-sided alpha value of 0.05.

## 3. Results

### 3.1. Demographic and Clinical Characteristics

In this study, 71 HCC patients undergoing combined targeted therapy and immunotherapy were recruited. The median age of the participants was 57 years, with a male predominance (58 men vs. 13 women). The etiology of HCC was diverse: 50 cases were attributed to chronic hepatitis B, 4 to hepatitis C, and 7 to alcoholic hepatitis, while 10 patients had no identifiable underlying cause. According to the Barcelona Clinic Liver Cancer (BCLC) staging, the cohort comprised 19 patients in stage B, 34 patients in stage C (PVTT), and 18 patients in stage C (M). The enrolled patients generally exhibited good liver function compensation, with 34 in Child–Pugh stage A and 37 in stage B. Regarding PD-1 immune checkpoint inhibitor therapy, 33 patients were treated with sindilizumab, 25 with camrelizumab, 9 with tislelizumab, and 4 with toripalimab. In terms of targeted therapy selection, lenvatinib was the most commonly used drug as it was administered to 57 patients, while 12 received sorafenib and only 2 were treated with bevacizumab. Concurrently, we conducted a comparison of the baseline characteristics between two patient subgroups and discovered that the only discernible difference lay in the choice of targeted therapy. Comprehensive information can be found in Table 1.

The efficacy of treatment from the 71 enrolled patients was assessed using mRECIST 1.1. The results revealed that 1 patient attained a complete response (CR), 12 patients achieved a partial response (PR), 31 patients exhibited stable disease (SD), and 20 patients had progressive disease (PD). Regrettably, seven patients lacked conclusive imaging evidence, rendering efficacy evaluation impossible. Detailed data are presented in Table A1.

In this study, the incidence of irAEs among patients was determined through meticulous clinical follow-up, and their severity was systematically evaluated using the CTCAE. Differential diagnosis revealed that 34 out of the 71 patients enrolled in the study developed immune-mediated adverse reactions. The most frequently observed manifestations included rashes in 13 patients, fever in 17 patients, hypertension in 4 patients, hepatitis in 4 patients, thyroiditis in 2 patients, hyperthyroidism in 2 patients, hypothyroidism in 5 patients, hypophysitis in 1 patient, diarrhea/colitis in 7 patients, and pneumonia in 1 patient. Notably, 10 patients experienced grade 3–4 adverse reactions. Specifically, there were four cases of severe rash, two cases of high-grade fever, three cases of severe hepatitis, and one case of severe colitis. Comprehensive data related to these immune-related adverse reactions, including their specific manifestations, grades, and outcomes, are detailed in Table 2.

Regarding the onset time of irAEs, fever and rash were found to occur relatively early, with median times of 1.1 weeks and 1.7 weeks, respectively. One case of hypophysitis in which the adverse reaction manifested at 55.9 weeks after treatment was recorded. At the time of data analysis, this particular reaction remained unresolved; thus, its specific duration remains undisclosed (refer to Figure 1 for additional information).

### 3.2. Analysis of Plasma Proteomics Characteristics of irAEs

Horizontal analysis of protein differences detected using Olink between patients with and without irAEs at the same time point revealed the most pronounced disparities in serum proteomics three days after receiving a combination of targeted therapy and immunotherapy. These differences predominantly encompassed IL-10, CXCL10, CXCL11, and LIF-R, as depicted in Figure 2a,b. Furthermore, the Kyoto Encyclopedia of Genes and Genomes (KEGG) pathway enrichment analysis of these differentially expressed proteins indicated that they were predominantly involved in the interaction between cytokines and cytokine receptors, as illustrated in Figure 2c.

The state observed at the onset of irAEs is highly representative. Capitalizing on the strengths of a longitudinal cohort design, our study prospectively collected matched plasma samples from the same patients during the occurrence of irAEs, at baseline before treatment initiation, and following the offset phase of irAEs for subsequent Olink analysis. By comparing the plasma proteomic profiles during irAEs with those obtained at the baseline, we identified marked differences, particularly in the upregulated expression of chemokines, including CXCL-9, CXCL-10, CXCL-11, and CCL3 (Figure 3a,b). Additionally, the KEGG pathway enrichment analysis uncovered significant alterations in the interactions between cytokines and cytokine receptors, further highlighting the dynamic nature of these immune responses (Figure 3c).

### 3.3. The Monocyte Proportion Increases during the Progression of irAEs

Our study included PBMC samples from two patients with immune-related hepatitis; they were collected before the administration of targeted immunotherapy, during the occurrence of irAEs, and after the resolution of irAEs. Following quality control filtering, a total of 37,586 cells were obtained and classified into nine cell populations based on characteristic genes, including CD8+ T cells, CD4+ T cells, NKT cells, NK cells, B cells, CD14+ monocytes, CD16+ monocytes, cDC, and Mega (Figure 4a,b). By mapping the differential markers from OLINK to the single-cell sequencing data, only EIF4EBP1 (4E-BP1), CCL3, and CD40 were detectable in the transcriptome data. Notably, CCL3 was specifically expressed in CD14+ monocytes at the transcriptional level. Its expression was low before targeted immunotherapy but increased in CD14+ monocytes during irAEs and subsequently decreased after the resolution of irAEs (Figure 4c). Furthermore, an analysis of the proportional trends of PBMC subpopulations at different time points revealed a notable increase in the proportion of CD14+ monocytes during irAEs. This trend persisted even after the offset phase of irAEs, with a further expansion in the proportion (Figure 4d).

### 3.4. CD14-MC-CCL3 during the Occurrence of irAEs

Based on the previously mentioned findings, we observed a correlation between the emergence of irAEs and CD14+ monocytes. Further characterization of distinct cell subpopulations led to the identification of four subsets within the CD14+ monocytes: CD14-MC-CCL3, CD14-MC-HLA-DPB1, CD14-MC-VCAN, and CD14-MC-PPBP. Specifically, CD14-MC-CCL3 is marked by the expression of IL1B, CCL3, and CXCL8; CD14-MC-HLA-DPB1 is distinguished by the expression of HLA-DPB1, HLA-DPA1, and HLA-DRB1; CD14-MC-VCAN exhibits high levels of VCAN, HMGB2, and FCN1; CD14-MC-PPBP is hypothesized to represent a distinct cellular state in which monocytes adhere to platelets, as demonstrated by the robust PPBP expression. Comprehensive gene expression profiles for other cell subpopulations are presented in Figure A1.

During the occurrence of irAEs, notable alterations were observed in the functional profile of CD14+ monocytes. Prior to the administration of targeted immunotherapy, the CD14-MC-HLA-DPB1 subtype predominantly comprised the highest proportion of CD14+ monocytes circulating in the peripheral blood of patients. However, upon the development of irAEs, there was a marked increase in the overall proportion of CD14+ monocytes in the peripheral blood, with the CD14-MC-CCL3 subpopulation emerging as the dominant subset. Following the offset phase of these immune-related reactions, a shift was observed toward an increased representation of the CD14-MC-VCAN subtype (Figure 5b).

The CD14-MC-CCL3 subpopulation specifically plays a pivotal role in innate immune responses, cytokine reactions, inflammatory reactions, and cytokine-mediated signaling pathways, highlighting its robust chemotactic and pro-inflammatory capabilities, as depicted in Figure 5a,c. To quantitatively evaluate the chemotactic ability of CD14+ monocytes at various time points, a monocyte chemotactic capacity score was established. Notably, during the manifestation of irAEs, there was a significant increase in the chemotactic capacity score of CD14+ monocytes circulating in the peripheral blood of patients, as shown in Figure 5d. Further investigation into intercellular communication uncovered that the cell subpopulation exhibiting cytotoxic function serves as a critical source of monocyte chemotactic signals, as illustrated in Figure 5e.

### 3.5. The IFN-γ Signaling Pathway Promotes Monocyte Activation during the Occurrence of irAEs

IFN-γ, the sole member of the type II interferon family, plays a pivotal role in immune regulation. Signaling through IFN-γ can activate monocytes, inducing potent cytotoxic and pro-inflammatory responses. To investigate this pathway, we introduced an IFN-γ signaling pathway score to analyze its ligand–receptor interactions. Our observations revealed that the IFN-γ signaling pathway score of CD14+ monocytes in the peripheral blood of patients was markedly elevated during irAEs compared with those in both the pre-treatment and post-resolution phases (Figure 6a). Concurrently, CD8+ T cells emerged as significant contributors to the IFN-γ signaling pathway. Analysis of the IFN-γ signaling pathway scores in CD8+ T cells at various time points during irAEs showed significantly higher scores in patient blood samples obtained during the active phase of irAEs, as opposed to before and after treatment (Figure 6b). Furthermore, our analysis of intercellular communication clarified a notable regulatory relationship between the CD8-Tprol cell subpopulation’s IFN-γ production and the CD14-MC-CCL3 cell subpopulation through specific signaling pathways (Figure 6c). This interaction underscores the dynamic crosstalk between these immune cell subsets during irAEs.

## 4. Discussion

This study focuses on the immunological characteristics of irAEs, with rash and fever being the most common, which is consistent with previous research [16,17,18]. IL-6 has a wide range of sources in peripheral blood and is often associated with inflammatory states [19]. Previous studies have reported associations between IL-6 and colitis, dermatitis, and arthritis in the context of irAEs [20,21,22]. However, due to the lack of specificity of IL-6, it is not sufficient as an independent indicator for predicting irAEs.

We further explored the proteomic characteristics of cytokines in the plasma of patients with irAEs using the Olink technology. We analyzed the proteomic differences between two groups of individuals with and without irAEs at the same time point and found that the differences were most significant three days after treatment. Subsequently, our results indicated that patients with irAEs exhibited elevated levels of chemokines such as CXCL9, CXCL10, CXCL11, and CCL3 in their peripheral blood during the occurrence of the adverse events. Current research suggests that the CXCL9, CXCL10, and CXCL11/CXCR3 axis regulates the migration, differentiation, and activation of immune cells, thereby inhibiting tumor development [23,24,25,26]. However, in our study, we did not observe superior outcomes in patients with irAEs compared with those without. The study of biological markers for irAEs can further stratify patient populations and guide clinical treatment plans. Currently, elevated levels of IL-6, CXCL5, CXCL9, and CXCL10 after treatment are associated with irAEs [27,28,29]. These findings are consistent with our results, but in our study, these chemokines were often elevated at the time of the adverse event, which occurred later. Therefore, we believe that proteomic indicators based on plasma samples collected three days after treatment may have predictive value for the occurrence of irAEs.

To validate the proteomics results, we attempted to map the differential indicators from Olink to transcriptome data to explore the source of the plasma differential indicators. However, unfortunately, we only detected gene expression corresponding to the expression of 4E-BP1, CCL3, and CD40 among the differential proteins at the transcriptional level of PBMCs. Among them, the CCL3 gene is enriched in CD14+ monocytes and transiently upregulated during irAEs. We further explored this using single-cell transcriptome data. In irAEs, our results showed an increase in the proportion of CD14+ monocytes, but the proportion did not return to normal after the resolution of the adverse event as we had initially hypothesized. Furthermore, we identified a CD14+ monocyte subset with strong chemotactic and inflammatory functions: CD14-MC-CCL3. This subset has been reported in previous immunological studies on knee osteoarthritis and Takayasu’s arteritis [30,31,32]. A functionally similar cell subset has also been annotated in a single-cell sequencing study on COVID-19 and is potentially associated with the cytokine storm in severe COVID-19 patients [33]. This, to some extent, validates our scientific hypothesis that irAEs are a type of cytokine storm involving multiple cytokines. Subsequently, we further analyzed the distribution and functional characteristics of four CD14+ monocyte subsets. Among them, the expression proportion of CD14-MC-CCL3 was significantly increased in peripheral blood samples during the occurrence of irAEs, whereas it remained at a lower proportion at baseline and after the resolution of irAEs. Meanwhile, we noticed that the proportion of CD14+ monocytes did not decrease after the resolution of irAEs. Further analysis of their functional status revealed that the CD14-MC-VCAN subset was the dominant CD14+ monocyte population after the resolution of irAEs. VCAN plays an important role in inflammatory responses, regulating leukocyte migration and activation and participating in the initiation and resolution of inflammation [34,35]. Similarly, VCAN is also an important indicator of the activation of myeloid-derived suppressor cells (MDSCs) [36].

To better elucidate the underlying mechanism of irAEs, we further analyzed the chemotactic function of CD14+ monocytes. We referenced gene sets related to monocyte chemotactic pathways to construct a monocyte chemotactic score [37,38]. Analysis at different time points revealed that the chemotactic score of CD14+ monocytes was significantly higher during irAEs than at the baseline and after the resolution of irAEs. Current research suggests a strong correlation between monocyte chemotaxis and the occurrence of inflammation [39,40]. This result further supports the association between the occurrence of irAEs and monocytes from a different perspective. Unexpectedly, we also found that the CD14-MC-PPBP subset exhibited strong chemotactic abilities, which, based on previous studies [41,42], may have been related to vascular endothelial damage in patients. The results of the cell–cell communication analysis identified unilateral chemotaxis of multiple cytotoxic cell subsets toward monocytes.

Thus far, our work has identified a group of monocytes that are highly correlated with the occurrence of irAEs. In further analysis, we attempted to explore the possible reasons for monocyte activation using transcriptome data. Previous studies have demonstrated that the IFN-γ signaling pathway is an important route for monocyte activation [43,44,45]. Similarly, a study published in 2021 used a mouse model to show that the occurrence of irAEs during immunotherapy is highly correlated with the activation of monocytes via the IFN-γ signaling pathway [46]. Therefore, we referenced previous research on the scoring of the IFN-γ signaling pathway and compiled relevant genes for subsequent data analysis [46,47,48]. During the occurrence of irAEs, the score of the IFN-γ signaling pathway in CD8+ T cells increased, which may have been related to the activation of the IFN-γ signaling pathway in CD8+ T cells after anti-PD-1 treatment, as suggested by previous studies [49,50,51]. Further analysis of the score of the IFN-γ signaling pathway in CD14+ monocytes revealed that it was significantly higher during irAEs than at the baseline and after the resolution of irAEs. Among the CD14+ monocyte subsets, the score of the IFN-γ signaling pathway in the CD14-MC-CCL3 subset was significantly higher than that in the CD14-MC-HLA-DPB1, CD14-MC-VCAN, and CD14-MC-PPBP subsets during irAEs. This result further confirms the correlation between the CD14-MC-CCL3 subset and irAEs. In subsequent analysis, we further validated the important role of the IFN-γ signaling pathway in communication between CD8+ T cells and the CD14-MC-CCL3 subset through cell–cell communication analysis.

## 5. Conclusions

In conclusion, this study delves into the proteomics of peripheral blood and the characteristics of single-cell transcriptomics linked to irAEs in patients with hepatocellular carcinoma undergoing PD-1 inhibition combination therapy. Notably, substantial alterations in plasma proteins were detected within just three days of therapy commencement, suggesting their possible role in anticipating long-term irAEs. Nevertheless, owing to a limited sample size, we could not formulate a predictive model. This study reveals an elevation in the ratio of CD14-MC-CCL3 monocytes during irAEs, affirming their pronounced proinflammatory activity. However, it is worth noting that, due to the confined sample size, a deeper investigation into the precise mechanism is warranted. Moreover, by analyzing intercellular interactions, we probed the likely mechanism that involves the activation of the IFN-γ signaling pathway in CD14-MC-CCL3 cells.

## Figures and Tables

**Figure 1 biomedicines-12-01140-f001:**
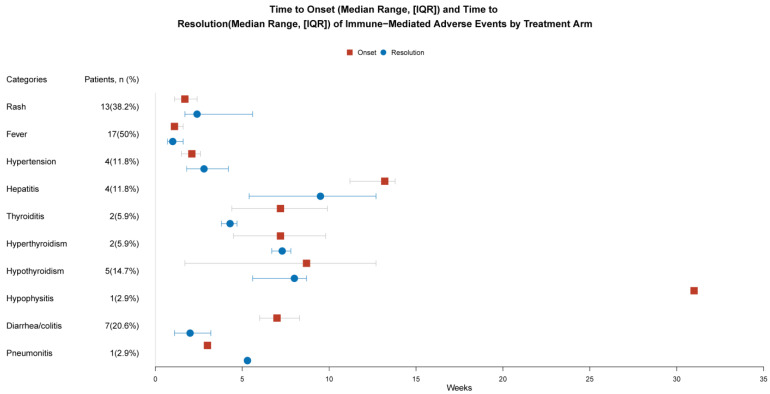
The time of occurrence and duration of immune-related adverse events.

**Figure 2 biomedicines-12-01140-f002:**
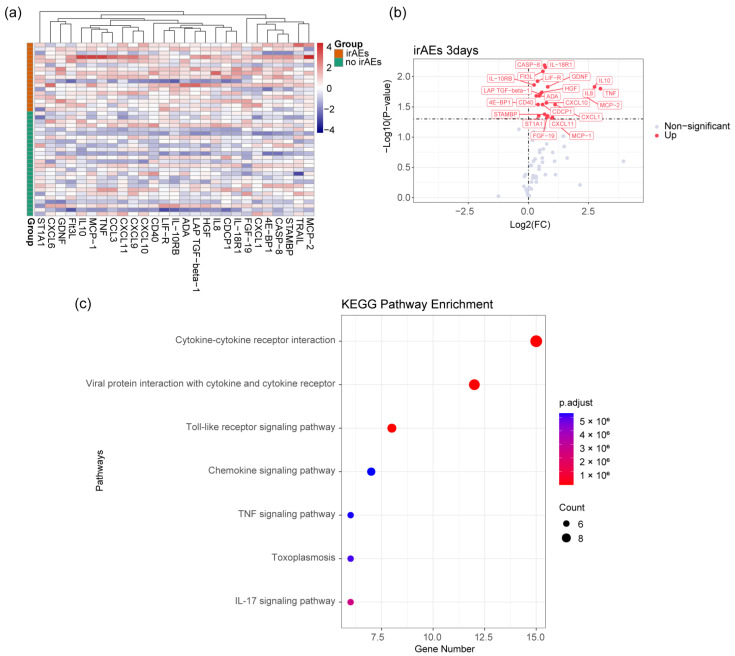
Proteomic differential analysis in irAEs three days after treatment. (**a**) Heatmap of plasma differential proteins in patients with irAEs three days after treatment. (**b**) Volcano plot of plasma differential proteins in patients with irAEs three days after treatment. (**c**) KEGG pathway enrichment analysis of plasma differential proteins in patients with irAEs three days after treatment.

**Figure 3 biomedicines-12-01140-f003:**
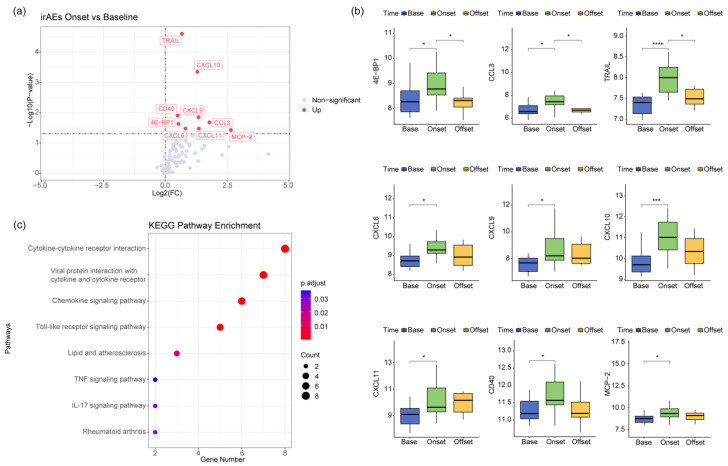
Proteomic differential analysis at the onset of irAEs. (**a**) Volcano plot of plasma differential proteins at the onset of irAEs. (**b**) Temporal trends of differential markers during irAEs. (**c**) KEGG pathway enrichment analysis of differential plasma proteins in patients at the onset of irAEs. ****: ≤0.0001; ***: ≤0.001; *: ≤0.05.

**Figure 4 biomedicines-12-01140-f004:**
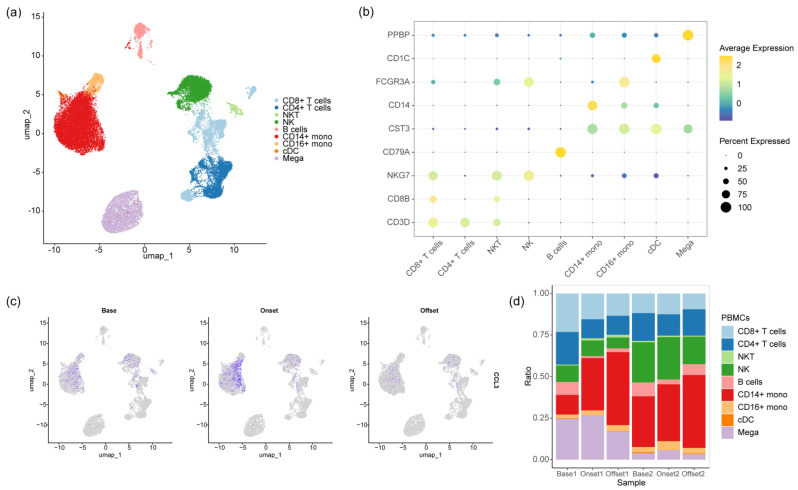
Monocytes are associated with the occurrence of irAEs. (**a**) Annotation of cell populations in single-cell sequencing. (**b**) Distribution of gene expression across cell populations. (**c**) Transient elevation of CCL3 expression in monocytes during the occurrence of irAEs. (**d**) Changes in cell proportions within PBMCs during the development of irAEs.

**Figure 5 biomedicines-12-01140-f005:**
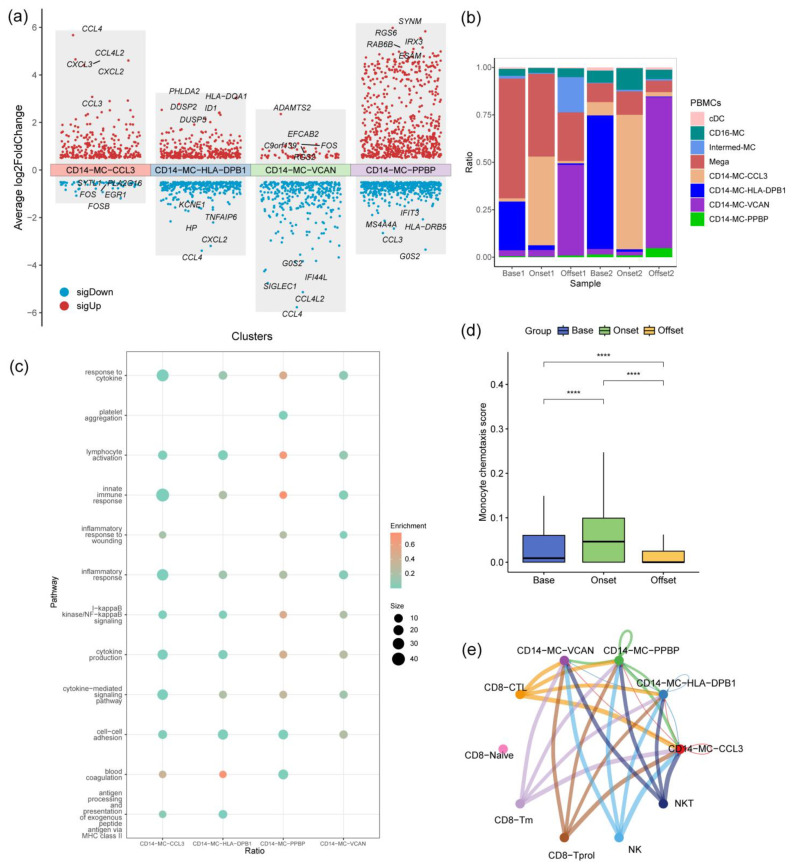
Functional analysis of monocyte subpopulations. (**a**) Differential gene expression analysis of monocyte subpopulations. (**b**) The proportion of myeloid cells changes with the progression of irAEs. (**c**) Enrichment analysis of characteristic gene pathways in monocyte subpopulations. (**d**) Changes in the monocyte chemotaxis score with the progression of irAEs. (**e**) Analysis of CCL signaling pathway communication between monocytes and cytotoxic cells. ****: ≤0.0001.

**Figure 6 biomedicines-12-01140-f006:**
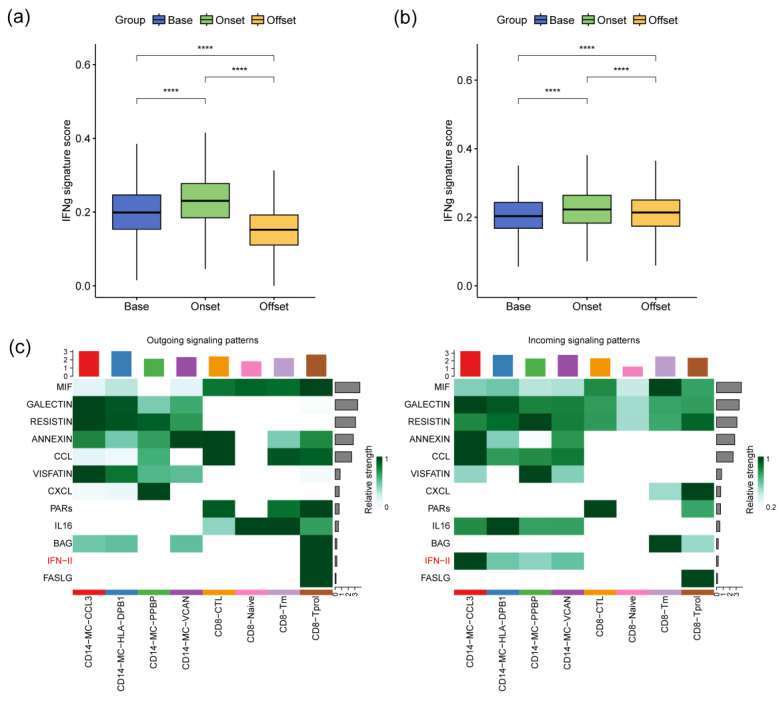
Correlation analysis between the IFN-γ signaling pathway and irAEs. (**a**) Changes in the IFN-γ signaling pathway score of CD14+ monocytes with the progression of irAEs. (**b**) Changes in the IFN-γ signaling pathway score of CD8+ T cells with the progression of irAEs. (**c**) Analysis of cellular communication between CD14+ monocyte subpopulations and CD8+ T cell subpopulations. ****: ≤0.0001.

**Table 1 biomedicines-12-01140-t001:** Baseline characteristics of the patients enrolled in the study.

Characteristics	No irAEs (*n* = 37)	irAEs (*n* = 34)	*p*
Age (years)	55.51 ± 12.24	59.32 ± 10.18	0.157
Sex			0.876
Male	30 (81.1%)	29 (85.3%)	
Female	7 (18.9%)	5 (14.7%)	
Basics of hepatitis			0.680
Viral hepatitis B	30 (81.1%)	27 (79.4%)	
Viral hepatitis C	3 (8.1%)	1 (2.9%)	
Alcoholic hepatitis	3 (8.1%)	4 (11.8%)	
Without basics of hepatitis	1 (2.7%)	2 (5.9%)	
BCLC stage			0.390
B	12 (32.4%)	7 (20.6%)	
C (PVTT)	15 (40.5%)	19 (55.9%)	
C (M)	10 (27.1%)	8 (23.5%)	
Child–Pugh stage			0.710
A	19 (51.4%)	15 (44.1%)	
B	18 (48.6%)	19 (55.9%)	
AFP			1
<400 ng/mL	26 (70.3%)	23 (67.6%)	
≥400 ng/mL	11 (29.7%)	11 (32.4%)	
MELD	9.5 (7, 11.5)	8.5 (7, 11.5)	
Immunotherapy			0.287
Sintilimab	14 (37.8%)	19 (55.9%)	
Camrelizumab	14 (37.8%)	11 (32.4%)	
Tislelizumab	7 (18.9%)	2 (5.9%)	
Toripalimab	2 (5.4%)	2 (5.9%)	
Combination targeted treatment			<0.01
Lenvatinib	25 (67.6%)	32 (94.1%)	
Sorafenib	10 (27.0%)	2 (5.9%)	
Bevacizumab	2 (5.4%)	0	
Combined local–regional treatment			0.410
TACE	8 (21.6%)	3 (8.8%)	
Tumor ablation	6 (16.2%)	4 (11.8%)	
Radiation therapy	1 (2.7%)	1 (2.9%)	
No	22 (59.5%)	26 (76.5%)	
Objective response rate, ^a^	7 (18.9%)	6 (17.6%)	1
Disease control rate, ^b^	24 (64.9%)	20 (58.8%)	0.780

Abbreviations: AFP, alpha fetoprotein; BCLC, Barcelona Clinic Liver Cancer; MELD, Model for End-Stage Liver Disease; TACE, transcatheter arterial chemoembolization. ^a^ Defined as complete response + partial response. ^b^ Defined as complete response + partial response + stable disease + non-complete response/non-progressive disease.

**Table 2 biomedicines-12-01140-t002:** Occurrence of immune-related adverse events.

	Patients with irAEs (*n* = 34)	
	Any Grade	Grade 3–4
Rash	13 (38.2%)	4 (11.8%)
Fever	17 (50%)	2 (5.9%)
Hypertension	4 (11.8%)	0
Hepatitis	4 (11.8%)	3 (8.8%)
Thyroiditis	2 (5.9%)	0
Hyperthyroidism	2 (5.9%)	0
Hypothyroidism	5 (14.7%)	0
Hypophysitis	1 (2.9%)	0
Diarrhea/colitis	7 (20.6%)	1 (2.9%)
Pneumonitis	1 (2.9%)	0

Abbreviations: irAEs, immune-related adverse events.

## Data Availability

The data used to support the findings of this study are available from the corresponding author upon reasonable request.

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
