# Peer review of "Activation of CD14+ Monocytes via the IFN-γ Signaling Pathway Is Associated with Immune-Related Adverse Events in Hepatocellular Carcinoma Patients Receiving PD-1 Inhibition Combination Therapy"

_biomedicines, 2024, doi:10.3390/biomedicines12061140_

Round 1

Reviewer 1 Report

Comments and Suggestions for Authors

The findings of the study are interesting. Defining the mechanisms of inflammation that are up-regulated in patients that experiment IRAE's and the  mechanisms that mediate resolution of inflammation are important to define novel biomarkers and novel therapeutic targets to reduce side effects to IC therapy. The study suggests that monocyte responses are induced that however in a patient with chronic tumor burden perpetuate cancer-driving inflammation instead of achieving anti-tumor effector responses. It would be interesting to compare those signatures to monocyte signatures in patients that show efficacy to IC therapy. 

The soundness of the current manuscript is unfortunately limited due to the small sample size of n=2 patients. In addition patients with HCC are a very heterogenous population. Therefore  a representative patients population is required to draw meaningful conclusions of scientific soundness.

Comments on the Quality of English Language

Minor editing of English language required.

Author Response

We are deeply honored to have received your invaluable advice and guidance. We firmly believe that this input will significantly enhance the quality of our work and prove to be an asset of immense strategic value for our future undertakings.

First and foremost, we profoundly recognize the foresightfulness of your scientific perspective, particularly the emphasis on exploring the relationship between inflammation and therapeutic efficacy in immunotherapy as a pivotal direction in tumor immunology. Aligning with your vision, our research team intends to delve deeper into this critical scientific domain.

In the course of our study, we encountered substantial challenges due to the unpredictable nature of the occurrence of immune-related adverse events (irAEs). This uncertainty complicated our follow-up efforts and sample collection endeavors. Consequently, our single-cell sequencing analysis was limited to six samples, collected from three different time points of two representative patients. We acknowledge this as one of the primary constraints of our study. The aim of analyzing this dataset is to investigate certain observations derived from the Olink results and to elucidate their potential mechanisms. Grateful for your insightful suggestions, we are committed to expanding our sample collection to further substantiate our findings in future investigations.

Meanwhile, we deeply appreciate your invaluable language revision suggestions. Having refined the article based on your feedback, we are confident that our paper will now be more comprehensible for readers.

Finally, we extend our heartfelt gratitude to you for finding the time, amidst your demanding schedule, to offer meticulous, scientifically sound, and insightful recommendations for our study. May you enjoy excellent health and attain professional fulfillment.

Reviewer 2 Report

Comments and Suggestions for Authors

The authors investigated the sources of Immune-related adverse events (irAEs) in HCC patients receiving PD-1 inhibition combination therapy.

The research is interesting but I have some questions and comments. 

1- Did all patients receive similar set of drugs?

2- Regarding high complexity in human samples and genetic differences, how authors rely on the results of single cell sequencing from 1 individual in each group? 

3- In the graphs the term "offset" was frequently used, but authors never used this term in the text. 

4- The interpretation of the KEGG pathway analysis was not mentioned in the text. Also, how the results are integrated with the single cell transcriptomics?

5- Line 180, "The efficacy of the 71 enrolled patients was assessed using the mRECIST 1.1." Did author mean the efficacy of [treatment] from the 71 enrolled patients?

6- line 357-359, "the expression proportion of CD14-MC-CCL3 was significantly increased 357 in peripheral blood samples during the occurrence of irAEs, while it decreased at baseline 358 and after the resolution of irAEs." 

The expression did not decrease at the baseline, the abundance was low indeed. 

7- Line 285, "the CD4+ Tem cell subset", the T effector memory (TEM) is meant? 

8- For single cell transcriptomics, instead of sample1 and sample2, it's better to show which one has or does not have the irAEs.

Comments on the Quality of English Language

The overall quality of the manuscript is good, but there are some punctuation errors. 

Author Response

Dear expert, we sincerely appreciate your professional and meticulous guidance. Your suggestions have undoubtedly refined our work, enhancing the rigor and scientific validity of our paper. Furthermore, they serve as a crucial reference for our future scientific endeavors. We deeply admire your rigorous and pragmatic approach to science.

1- Did all patients receive similar set of drugs?

Immunotherapy-based combination therapy holds a prominent position in the current landscape of drug treatments for liver cancer. In our study, we exclusively enrolled patients undergoing this type of therapy. Incorporating your invaluable feedback, we have highlighted the following key points in our manuscript: For the administration of immunotherapy drugs, Sintilimab, Tislelizumab, and Camrelizumab were given at fixed doses of 200mg each, while Toripalimab was ad-ministered at a slightly higher dose of 240mg. All immunotherapies followed a three-week treatment cycle. In terms of targeted therapy, Sorafenib was prescribed at a daily dose of 400mg. Lenvatinib dosage ranged from eight to 12mg daily, adjusted ac-cording to the patient's body weight. Bevacizumab, however, was dosed at 15mg per kilogram of body weight per day.

It is worth emphasizing that the diagnosis and differentiation of irAEs during combination therapy constitute the cornerstone of our clinical practice. This crucial task is undertaken by seasoned clinicians within our team who possess extensive clinical experience.

Initially, we differentiate irAEs primarily based on the timing and symptoms of adverse reactions.

In certain cases, we opt to discontinue ICI treatment to ascertain whether adverse drug reactions are indeed indicative of irAEs.

Furthermore, during the experimental design phase, we consult previous research and conduct dynamic monitoring of patients' IL-6 levels. This serves as a valuable diagnostic tool, providing crucial insights into the onset of irAEs.

2- Regarding high complexity in human samples and genetic differences, how authors rely on the results of single cell sequencing from 1 individual in each group?

The uncertainty surrounding the occurrence time of irAEs complicates our clinical sample collection efforts. As a result, we carefully selected six samples from two representative patients across three distinct disease states to serve as the foundation for our single-cell transcriptome analysis. To derive our research findings, we performed a horizontal comparison of disparities at different time points. Additionally, the primary objective of this dataset is to delve deeper into our observations from the Olink data and unravel the potential mechanisms responsible for the temporary surge in plasma CCL3 levels.

3- In the graphs the term "offset" was frequently used, but authors never used this term in the text.

Your suggestion is invaluable. The original description had the potential to confuse readers, and we are genuinely grateful that you pointed out this issue. In our picture, we employed the term "offset" to depict the disease state following the cessation of irAEs. However, we neglected to mention this in the original text, which undoubtedly constituted a flaw in our writing. To rectify this, we have implemented corresponding adjustments to the original text.

4- The interpretation of the KEGG pathway analysis was not mentioned in the text. Also, how the results are integrated with the single cell transcriptomics?

Thank you for your valuable feedback. By introducing the KEGG results, our aim is to delve into the inner workings of Olink's differential indicators and gain insights into the cellular pathways involved in intercellular interactions through single-cell sequencing data. This analysis provides valuable clues for understanding these interactions. Our interpretation of the KEGG analysis findings is as follows: Additionally, KEGG pathway enrichment analysis uncovered significant alterations in the interactions between cytokines and cytokine receptors, further highlighting the dynamic nature of these immune responses.

5- Line 180, "The efficacy of the 71 enrolled patients was assessed using the mRECIST 1.1." Did author mean the efficacy of [treatment] from the 71 enrolled patients?

Thank you for your meticulous guidance. Indeed, this paragraph is describing the tumor treatment efficacy of the enrolled patients. To avoid ambiguity, we have further revised it to: The efficacy of treatment from the 71 enrolled patients was assessed using the mRECIST 1.1.

6- line 357-359, "the expression proportion of CD14-MC-CCL3 was significantly increased 357 in peripheral blood samples during the occurrence of irAEs, while it decreased at baseline 358 and after the resolution of irAEs."

The expression did not decrease at the baseline, the abundance was low indeed.

We are deeply grateful for your meticulous review of our manuscript. Your feedback carries significant weight, and we recognize that the initial description of the results could lead to misinterpretation by readers. After careful deliberation, we have further revised it to: Among them, the expression proportion of CD14-MC-CCL3 was significantly increased in peripheral blood samples during the occurrence of irAEs, whereas it remained at a lower proportion at baseline and after the resolution of irAEs.

7- Line 285, "the CD4+ Tem cell subset", the T effector memory (TEM) is meant?

Thank you for identifying the shortcomings in our paper. You were right in noting that our aim was to elaborate on the function of T effector memory. After integrating your insightful recommendations with the paper's core focus, we have opted to omit the controversial sentence, allowing us to concentrate on our primary message.

8- For single cell transcriptomics, instead of sample1 and sample2, it's better to show which one has or does not have the irAEs.

Thank you for your invaluable feedback. In our investigation, we gathered single-cell sequencing samples from two individuals who experienced immune-related adverse events (irAEs) at three distinct time intervals. Our analytical approach involved comparing the samples from these patients before, during, and after the occurrence of irAEs, enabling us to gain a comprehensive understanding of the immune response throughout this process.

Meanwhile, we deeply appreciate your invaluable language revision suggestions. Having refined the article based on your feedback, we are confident that our paper will now be more comprehensible for readers.

Finally, we extend our heartfelt gratitude to you for finding the time, amidst your demanding schedule, to offer meticulous, scientifically sound, and insightful recommendations for our study. May you enjoy excellent health and attain professional fulfillment.

Reviewer 3 Report

Comments and Suggestions for Authors

Authors showed the results of immune phenotype using proteomic profiling and single-cell RNA sequencing in HCC patients treated with ICI-containing regimens. As the results, authors showed that activation of CD14-monocytes via IFNg signaling pathway is associated with irAEs. This study was interesting and valuable. But several issues remained to be addressed.

1. In present study, combined immunotherapy was performed. Therefore, the symptoms might be confused. Hypertensin could be induced by VEGF blocking. TKIs also induced diarrhea. Authors should clarify the definition of irAEs. Confirmation of histological study for irAEs might be preferred.

2. The combined regimens were different between irAEs and no-irAEs. It should be justified. 

3. The limitations including the limited number of cases should be described.  

Comments on the Quality of English Language

Minor grammatical errors or miss-spellings were found.

Author Response

We are deeply honored that an expert with extensive clinical experience has reviewed our manuscript. Your invaluable strategic insights and professional guidance have significantly elevated the quality of our current study and have sparked crucial ideas for our future research endeavors.

1.In present study, combined immunotherapy was performed. Therefore, the symptoms might be confused. Hypertensin could be induced by VEGF blocking. TKIs also induced diarrhea. Authors should clarify the definition of immune-related adverse events (irAEs). Confirmation of histological study for irAEs might be preferred.

As you rightly pointed out, the differential diagnosis of irAEs poses a significant challenge in clinical practice. We differentiate these events primarily based on the following considerations:

â‘ .Firstly, symptoms serve as a crucial reference point, with the involvement of multiple bodily systems being a hallmark of irAEs. You accurately observed that hypertension and diarrhea can manifest as adverse reactions to various medications. Consequently, in this study, we have excluded patients presenting solely with hypertension or diarrhea. It bears mentioning that the diagnosis and differentiation of irAEs are undertaken by seasoned physicians within our clinical team.

â‘¡.In certain cases, we opt to discontinue immune checkpoint inhibitors (ICIs) to ascertain whether the adverse reaction is indeed an irAE.

â‘¢.Furthermore, drawing from previous research, we have incorporated dynamic monitoring of patients' IL-6 levels into our experimental design. This serves as an indicator suggestive of the onset of irAEs, providing us with a valuable auxiliary diagnostic tool.

â‘£.I am deeply grateful for your insightful advice. The significance of tissue in situ research cannot be overstated. We are diligently collecting samples for this endeavor, which will undoubtedly shape the direction of our future research endeavors.

2.The combined regimens were different between irAEs and no-irAEs. It should be justified.

Thank you very much for your valuable advice. This limitation is indeed inherent in our work as an observational study, where achieving consistency in drug selection among groups poses significant challenges. The evolving landscape of liver cancer treatment, continually refreshed by promising clinical studies introducing new combination therapies, further complicates the picture. In devising treatment plans, we carefully consider the patient's disease status, comorbidities, and economic factors to tailor the most suitable approach. Consequently, our study provides an objective portrayal of the real-world application of combination therapies.

It's noteworthy that TKI therapies predominate in our patients' treatment regimens. Variations between the two study groups primarily stem from differences in TKI selection. However, following a meticulous assessment of irAEs, we have determined that these group disparities exert minimal influence on our ultimate findings.

The experimental strategy you proposed is highly valuable. Following our team discussion, we have embraced your recommendations and will incorporate them into our future endeavors. Moving forward, we are committed to conducting rigorous prospective studies, establishing well-defined control and experimental groups to delve deeper into the mechanisms underlying irAEs.

3.The limitations including the limited number of cases should be described.

Thank you for your constructive criticism and invaluable guidance. Indeed, the limited sample size represents a significant constraint in our research. We have revised the text to reflect this limitation: " However, it's worth noting that, due to the confined sample size, a deeper investigation into the precise mechanism is warranted " Following your insightful suggestion, we plan to enlarge our sample size in future studies to enhance the validity and generalizability of our findings.

Meanwhile, we deeply appreciate your invaluable language revision suggestions. Having refined the article based on your feedback, we are confident that our paper will now be more comprehensible for readers.

Finally, we sincerely thank you for providing a comprehensive and reliable evaluation of our work, as well as important guidance. We apologize for the limitations of our research and hope to gain your understanding. May you enjoy excellent health and attain professional fulfillment.

Reviewer 4 Report

Comments and Suggestions for Authors

A very nice paper. 

Adverse effects of ICI are nowadays a main concern and one of the clinical indications to stop ICI therapy. The paper has a good interest in the HCC setting since the majority of patients are selected for ICI based on radiology and clinical features.

The possibility of sampling peripheral blood and collecting information for clinical strategies is therefore a must-have, especially when this kind of methodology is available worldwide.

Suggestion for manuscript improvement: 

Of the 71 patients, how many had tumour biopsy? It would have extra value to correlate with histological features, even in a small subcohort, namely with a degree of differentiation.

The English language is ok. Pictures/Graphs are of good quality

Author Response

We are deeply honored to have received your invaluable advice and guidance. We firmly believe that this input will significantly enhance the quality of our work and prove to be an asset of immense strategic value for our future undertakings.

We are extremely grateful for your invaluable advice, which provides exceptional guidance. As you mentioned, the exploration of tumor sites has always been a significant research focus. Understanding the differences between tumor sites and peripheral areas will facilitate a deeper comprehension of the underlying mechanisms of tumorigenesis and progression. This is a crucial scientific issue. Regrettably, in our current workflow, we have not systematically collected clinical samples from the original tumor sites of patients.

We attach great importance to your precious guidance. In subsequent clinical work, we will systematically collect samples for relevant research, provided it complies with ethics and the wishes of the patients. We sincerely thank you for providing important research directions and ideas for our future scientific endeavors.

Finally, we extend our heartfelt gratitude to you for finding the time, amidst your demanding schedule, to offer meticulous, scientifically sound, and insightful recommendations for our study. May you enjoy excellent health and attain professional fulfillment.

Round 2

Reviewer 1 Report

Comments and Suggestions for Authors

As mentioned in first review and by the authors, the limited sample size limits the significance of the data.

Comments on the Quality of English Language

Minor editing of English required.

Author Response

We are immensely grateful to have the opportunity to refine our current work under your invaluable guidance and advice. Your insights have not only enriched our understanding but also provided crucial ideas for shaping our future experimental plans. We are confident that with your assistance, our scientific research will undergo significant enhancement.

As you pointed out, the scarcity of single-cell sequencing samples poses a challenge in our present study. To address this, we intend to collaborate with various centers in our future endeavors, aiming to clarify and expand upon our key discoveries in single-cell sequencing.

Regarding our methodology, our manuscript is structured into three primary sections. Initially, we present an unbiased depiction of the frequency and duration of irAEs within our study cohort. Following this, we detail the proteomics analysis we conducted using Olink technology on 181 plasma samples collected from 71 participants at different time intervals. Lastly, we delve into the single-cell sequencing analysis of PBMCs obtained from two exemplar patients at three distinct time points. Notably, we observed a temporary surge in CCL3 expression among monocytes during irAEs. We came across Llewellyn et al. (2021) exploration of liver cancer irAEs mechanisms in a mouse model, which underscores the pivotal role of the IFN-γ signaling pathway, referred to Document 46. In our work, we used human samples for validation. Based on the aforementioned results, we have made the decision to incorporate the findings from single-cell sequencing into our present manuscript, with the belief that it will provide valuable insights and inspiration to our peers.

We are deeply appreciative of your constructive feedback. Moving forward, we are committed to continuously gathering clinical samples and integrating them with tissue in situ results to further investigate our current observations.

Finally, we offer our deepest gratitude to you for squeezing time out of your busy schedule to provide meticulous, scientifically rigorous, and insightful recommendations for our study. We sincerely wish you excellent health and profound professional fulfillment.

Reviewer 2 Report

Comments and Suggestions for Authors

The manuscript has significantly improved, yet further revisions are required. 

1) A clear explanation is required on the calculation of scores, including monocytic chemotaxis and IFNγ signature, as well as the number of observations gathered. 

2) In Figure 6b, the base and offset appear significantly different with a p-value of less than 0.0001. This seems counterintuitive as they appear similar. Could you please verify the statistical analysis?

3) "Activation of CD14+ Monocytes via the IFN-γ Signaling Pathway" was deduced from indirect analysis and has not been substantiated by direct methods or through induction or inhibition strategies. Therefore, it seems premature to use this as a title.

Comments on the Quality of English Language

The English still needs revision.

For example, in abstract, line 23, there need a c conjunctive adverb before "The specific ..."

- line 310, "(Figure 6b).Furthermore,". There need a space after full stop.  

Author Response

We are immensely grateful to have the opportunity to refine our current work under your invaluable guidance and advice. Your insights have not only enriched our understanding but also provided crucial ideas for shaping our future experimental plans. We are confident that with your assistance, our scientific research will undergo significant enhancement.

1) A clear explanation is required on the calculation of scores, including monocytic chemotaxis and IFNγ signature, as well as the number of observations gathered.

Thank you very much for your valuable advice. We have provided a detailed description of the scoring calculation method as follows:

2.7. Defining cell state scores

We used the AUCell-R package to score the target gene sets. With reference to the relevant literature, we used MONOCYTE CHEMOTAXIS (GO:0002548) for the monocyte chemotaxis score and TYPE II INTERFERON-MEDIATED SIGNALING PATH-WAY (GO:0060333) for the IFN-γ signaling pathway score.

2) In Figure 6b, the base and offset appear significantly different with a p-value of less than 0.0001. This seems counterintuitive as they appear similar. Could you please verify the statistical analysis?

Thank you very much for your thoughtful advice. We have carefully reviewed the statistical results and can confirm their accuracy. In our scoring analysis, each individual cell served as the fundamental unit of measurement. Consequently, even minor variations can become magnified when dealing with a large number of cells, ultimately leading to notable differences in p-values.

3) "Activation of CD14+ Monocytes via the IFN-γ Signaling Pathway" was deduced from indirect analysis and has not been substantiated by direct methods or through induction or inhibition strategies. Therefore, it seems premature to use this as a title.

Thank you very much for your valuable advice. The rationale behind our paper's title stems primarily from Llewellyn et al.'s (2021) investigation into the mechanisms of liver cancer irAEs in a mouse model. This study, referenced in Document 46, emphasizes the crucial role of the IFN-γ signaling pathway. It served as a significant inspiration for our experimental design and data interpretation. Nevertheless, a limitation of Llewellyn's study is the lack of clinical sample validation. Hence, our research aims to bridge this gap by building upon previous findings and striving for meaningful outcomes. Additionally, we plan to broaden our sample pool and conduct additional in vitro validation studies. We are deeply honored to have the opportunity to share our scientific thinking with you, and we genuinely hope to receive your comprehension and backing.

We are thrilled to have received prompt and expert feedback from the MDPI editing team regarding language enhancement, which we strongly believe will elevate the quality of our manuscript significantly.

Finally, we offer our deepest gratitude to you for squeezing time out of your busy schedule to provide meticulous, scientifically rigorous, and insightful recommendations for our study. We sincerely wish you excellent health and profound professional fulfillment.

Reviewer 3 Report

Comments and Suggestions for Authors

Revised manuscript was well-addressed to the comments and well-written.

Author Response

We are immensely grateful for your guidance and assistance, which have greatly aided us in refining our manuscript and shaping crucial ideas for our upcoming scientific endeavors. Your dedication to finding time amidst your busy schedule to deliver precise, scientifically sound, and enlightened suggestions for our research is deeply appreciated. We sincerely wish you robust health and remarkable success in your esteemed career.